# Pulmonary and Systemic Immune Profiles Following Lung Volume Reduction Surgery and Allogeneic Mesenchymal Stromal Cell Treatment in Emphysema

**DOI:** 10.3390/cells13191636

**Published:** 2024-09-30

**Authors:** Li Jia, Na Li, Vincent van Unen, Jaap-Jan Zwaginga, Jerry Braun, Pieter S. Hiemstra, Frits Koning, P. Padmini S. J. Khedoe, Jan Stolk

**Affiliations:** 1Department of Immunology, Leiden University Medical Center (LUMC), 2333 Leiden, The Netherlands; l.jia@lumc.nl (L.J.);; 2Department of Pulmonology, PulmoScience Lab, Leiden University Medical Center (LUMC), 2333 Leiden, The Netherlandsj.stolk@lumc.nl (J.S.); 3State Key Laboratory for Diagnosis and Treatment of Severe Zoonotic Infectious Diseases, Key Laboratory of Zoonosis Research of the Ministry of Education, Institute of Zoonosis and College of Veterinary Medicine, Jilin University, Changchun 130012, China; 4Department of Hematology, Leiden University Medical Center, 2333 Leiden, The Netherlands; 5Department of Cardiothoracic Surgery, Leiden University Medical Center, 2333 Leiden, The Netherlands

**Keywords:** bone marrow-derived mesenchymal stromal cells (BM-MSCs), emphysema, lung volume reduction surgery (LVRS), mass cytometry by-time-of-flight (CyTOF), pulmonary immune profile, systemic immune profile

## Abstract

Emphysema in patients with chronic obstructive pulmonary disease (COPD) is characterized by progressive inflammation. Preclinical studies suggest that lung volume reduction surgery (LVRS) and mesenchymal stromal cell (MSC) treatment dampen inflammation. We investigated the effects of bone marrow-derived MSC (BM-MSC) and LVRS on circulating and pulmonary immune cell profiles in emphysema patients using mass cytometry. Blood and resected lung tissue were collected at the first LVRS (L1). Following 6–10 weeks of recovery, patients received a placebo or intravenous administration of 2 × 10^6^ cells/kg bodyweight BM-MSC (n = 5 and n = 9, resp.) in week 3 and 4 before the second LVRS (L2), where blood and lung tissue were collected. Irrespective of BM-MSC or placebo treatment, proportions of circulating lymphocytes including central memory CD4 regulatory, effector memory CD8 and γδ T cells were higher, whereas myeloid cell percentages were lower in L2 compared to L1. In resected lung tissue, proportions of Treg (*p* = 0.0067) and anti-inflammatory CD163^−^ macrophages (*p* = 0.0001) were increased in L2 compared to L1, while proportions of pro-inflammatory CD163^+^ macrophages were decreased (*p* = 0.0004). There were no effects of BM-MSC treatment on immune profiles in emphysema patients. However, we observed alterations in the circulating and pulmonary immune cells upon LVRS, suggesting the induction of anti-inflammatory responses potentially needed for repair processes.

## 1. Introduction

Chronic obstructive pulmonary disease (COPD) is the third cause of human mortality in the world [1], and one of the most prevalent chronic non-communicable diseases. Emphysema as a component of COPD is characterized by damaged alveolar tissue and enlargement of alveolar air spaces. This results in progressive reduction in gas exchange, air flow, exercise capacity and quality of life [2]. Until now, lung transplantation has been the only curative treatment for emphysema. Pharmacological therapies have limited effects on patient symptoms, and do not halt or reverse tissue damage. Lung volume reduction surgery (LVRS) is an option to reduce lung hyperinflation and improve lung function in patients with severe emphysema, but does not halt the progression of emphysema [3,4].

Mesenchymal stromal cells (MSCs) have demonstrated anti-inflammatory and immunomodulatory properties [5]. MSCs can be isolated from stromal tissues including the bone marrow (BM-MSC), umbilical cord and adipose tissue. MSCs have been applied as therapy in clinical trials in various inflammatory diseases, including steroid-resistant graft-versus-host disease [6], sepsis [7], Crohn’s disease [8], diabetes mellitus [9], ulcerative proctitis [10], and respiratory diseases [11,12,13]. In a non-placebo controlled pilot study, we observed a significant increase in pulmonary CD31 expression upon intravenous administration of autologous BM-MSC 4 weeks prior to LVRS for severe emphysema [13]. We also observed a 2-fold increase in CD3^+^ T and CD4^+^ T cells in alveolar septa in resected lung tissue, suggesting that the immune composition is altered [13]. Previous in vitro studies showed that MSC can alleviate airway inflammation by the downregulation of cycloocygenase-2 (COX-2) and COX-2-mediated prostaglandin E2 (PGE2) production in macrophages [12]. MSC treatment also induces reprogramming of monocytes and macrophages to an immunosuppressive interleukin-10 (IL-10) producing phenotype [7]. Other studies reported immunomodulatory capability of MSC through secretion of interferon-gamma (IFN-γ) by T cells [14], promoting host defense and immunomodulation by induction of regulatory T cells [15]. BM-MSCs are also suggested to promote repair processes [16] by the secretion of growth factors that stimulate epithelial and endothelial repair. MSC treatment therefore presents an attractive treatment option for (chronic) inflammatory diseases, with tissue damage, such as emphysema. Currently, the effects of MSC therapy on the systemic and pulmonary immune cell composition are unknown, as is the effect of LVRS.

Here, we investigated the effect of LVRS and allogeneic BM-MSC treatment or placebo on the composition of circulating and pulmonary lymphoid and myeloid immune cells in affected lung tissue of patients with severe emphysema using mass cytometry by time of flight (CyTOF).

## 2. Materials and Methods

### 2.1. Patient Selection

Both male and female COPD patients, with an age range between 44 and 65 years, who met inclusion criteria for GOLD stage III or IV and were suitable to undergo staged bilateral LVRS were selected for randomization. The patients underwent lung volume reduction surgery with the aim to remove the most severely affected emphysematous lung tissue with trapped air from both lungs. Included patients had upper-lobe emphysema [17]. The surgical procedures were performed in two separate sessions, one lung operated on each session. The presence of emphysema was confirmed by a radiologist who reviewed chest Computed Tomography (CT) images. Included patients had to have a gradient of emphysema severity towards the lung apex as assessed by CT-derived lung densitometry by Pulmo CMS software (version number 2.2.0, Medis, Leiden, The Netherlands). The study protocol was approved by the Central Committee on Research Involving Human Subjects (CCMO) of The Netherlands and all study participants provided written informed consent. The study was registered at ClinicalTrials.gov (accessed on 9 June 2021) under the number NCT04918706. Additional inclusion and exclusion criteria [18] for patient selection are provided in the online Appendix A.

### 2.2. Experimental Design of the Clinical Study

Patients (n = 14) underwent the first LVRS (L1), and both lung tissue and arterial blood samples were collected. After 6 to 10 weeks, patients were randomized to receive 2 courses of intravenous treatment with BM-MSC (n = 9) or a placebo (n = 5), 1 week apart. BM-MSCs from healthy controls were processed and expanded as described before [13,19,20]. Patients underwent the second LVRS after 4 weeks, where arterial blood samples and lung tissue were collected again (Figure 1). Data on age and body mass index (BMI) were collected at baseline. Lung function measurements, including forced expiratory volume in 1 s (FEV1) and diffusing capacity of the lungs for carbon monoxide (Kco), were measured at baseline and at 12 months after the second LVRS (L2) at follow-up. However, 1 patient in the control group and 1 patient in the BM-MSC group did not undergo the second lung function measurement. Residual volume (RV)/total lung capacity (TLC), CT-derived lung densitometry, ejection fraction, and end-diastolic volume were measured at baseline. The number of pack-years and smoking history were collected at baseline (Table 1).

Patients (n = 14) underwent the first lung volume reduction surgery (LVRS, L1), and both distal lung tissue and arterial blood samples were collected. After 6 to 10 weeks, patients were randomized to receive 2 intravenous injections with bone marrow-derived mesenchymal stromal cells (BM-MSCs) (n = 9) or a placebo (n = 5), 1 week apart. Patients underwent a second contralateral LVRS (L2) after 3 weeks, where arterial blood samples and lung tissues were collected again.

### 2.3. Tissue Processing

Freshly collected lung tissue from lung volume reduction surgery (LVRS) was processed as described before [21]. Peripheral blood mononuclear cells (PBMCs) were isolated from freshly drawn and heparin anticoagulated arterial blood using Ficoll-Paque^TM^ density gradient centrifugation, as described before [22].

### 2.4. Mass Cytometry Antibody Staining and Data Analysis

High-dimensional mass cytometry by time of flight (CyTOF) was used for the analysis of PBMC and mononuclear cells harvested upon the enzymatic digestion of resected lung tissue with simultaneous measurement of 78 cellular markers at single-cell resolution [21,22,23]. Procedures for mass cytometry antibody staining and data acquisition were performed as described before [24] and are described in detail in the online Appendix A. The experiments were measured in 9 batches, and a reference PBMC sample was included in each batch as a staining control and for normalization [21]. For the data analysis, all single, live CD45^+^ cells in each sample were gated using FlowJo software version 10.6 (Tree Star Inc. (Ashland, OR, USA)). The data from different experimental batches were normalized using the reference PBMC samples and data were analyzed in Cytosplore [21]. A principal component analysis (PCA) was conducted on PBMC and lung samples to analyze the frequencies of immune cell clusters identified by a t-distributed stochastic neighbor embedding (tSNE) analysis. The centroids representing each patient group were then plotted in the PCA.

Pseudotime analysis of dynamic heterogeneous cell populations in PBMC and lung samples was applied as previously published [25]. In brief, all myeloid cells derived from PBMC and lung samples were collected for a five-level HSNE analysis in Cytosplore with myeloid-related markers. Subsequently, we identified 13 phenotypical clusters among the PBMC and resected lung samples, which were analyzed in R. Downsampling was performed to 2000 cell events, after which Slingshot was applied to the transformed myeloid cell clusters. The pseudotime variable was estimated through simultaneous fitting of principal curves and was visualized as a color gradient on a diffusion map. The R code is freely available on GitHub (https://github.com/janinemelsen/Single-cell-analysis-flow-cytometry (accessed on 6 October 2023)) [25].

### 2.5. Statistical Analysis

Data are presented as individual values by symbols and lines. Group comparisons of lymphoid and myeloid cell populations were performed with the Mann–Whitney U-test for paired non-parametric data in GraphPad (9.0.1(151)). *p* values were adjusted for the Wilcoxon matched-pair signed rank test (FDR < 5%). Differences at *p* values < 0.05 were considered statistically significant.

## 3. Results

Clinical parameters were unaltered upon LVRS and BM-MSC/placebo treatment. Baseline characteristics of included patients are shown in Table 1. At the post-trial follow-up, no adverse events (AEs) were reported after LVRS and BM-MSC/placebo treatment. There were no significant differences in bodyweight and pulmonary function parameters FEV_1_ and K_co_ compared to baseline (Figure 2A).

### 3.1. Increased Lymphoid Cells with Decreased Myeloid Cells in the Blood of Emphysema Patients upon LVRS

To determine the effect of LVRS on the circulating immune compartment, we applied CyTOF on PBMC from emphysema patients. We first identified the major immune lineages (B cells, CD4 T cells, CD8 T cells, CD3^−^CD7^+^ cells, and myeloid cells) in blood samples obtained at the first LVRS (L1) and second LVRS (L2) (Appendix A). In the total immune population, proportions of CD4 and CD8 T cell populations were higher in L2 compared to L1, while the proportion of myeloid cells was decreased (CD4 T cells: *p* = 0.0067, CD8 T: *p* = 0.0017) (Figure 3A and Appendix A). In a more detailed analysis of the CD4 T (Figure 3B), CD8/γδ T (Figure 3C), and myeloid (Figure 3D) cell compartments, we observed that the increase in the CD4 T cell compartment was explained by an increase in the Foxp3^+^ regulatory T cells (Treg) (*p* = 0.0052) and CD45RA^−^CCR7^+^ central memory T cells (Tcm) and not by CD45RA^+^CCR7^+^ naïve T cells (Tn), or CD45RA^−^CCR7^−^ effector memory T cells (Tem) (Figure 3B and Appendix A).

In the CD8 T cell population, we observed significantly higher proportions of Tem (*p* = 0.0001) and to a lesser extent γδ T cells (*p* = 0.0245) in L2 compared to L1 (Figure 3C). The proportion of CD3^−^CD7^+^ innate lymphoid cells (ILCs)/NK cells, expressing CD45RA, CD56, and Granzyme B, was comparable between L1 and L2 (Appendix A). Finally, the statistically significant decrease in proportions of circulating myeloid cells (*p* = 0.0012) in blood obtained during L2 was explained by lower percentages of CD14^+^CD163^+/−^ classical monocytes (*p* = 0.0245) (Figure 3D and Appendix A).

We performed a principal component analysis (PCA) to determine the effect of BM-MSC vs. placebo treatment on frequencies of the identified circulating cell subpopulations (Appendix A). We could distinguish L1 from L2 blood samples; however, we did not observe changes related to BM-MSC vs. placebo treatment (Appendix A). Taken together, in the circulation, we observed increased proportions of lymphoid cells and lower proportions of myeloid cells, which was mainly associated with LVRS, whereas the effects of BM-MSC versus placebo treatment were comparable.

### 3.2. Increased CD206^+^CD163^−^ Macrophages with Decreased CD206^+^CD163^+^ Macrophages and Dendritic Cells in Resected Lung Tissue of Emphysema Patients Post-LVRS

CD45^+^ immune cells obtained during L1 and L2 from lung tissue were also subjected to a CyTOF analysis. Firstly, the major pulmonary immune lineages (B cells, CD4 T cells, CD8 T cells, CD3^−^CD7^+^ cells, and myeloid cells) were identified (Appendix A). There were no significant differences in the major immune cell populations between the first and second LVRS (Figure 4A), and myeloid cells were found to be predominant among all immune cells in resected lung tissue (Appendix A).

Upon studying the T lymphocyte population in more detail, proportions of CD4 Tem and CD8 Tem cells were found to be dominant compared to other subpopulations (e.g., Tn, Tcm, Treg, γδ T) (Appendix A). Proportions of Treg (*p* = 0.0067) and CD4 Tn cells (*p* = 0.0107) were higher in L2 compared to L1 in the lungs (Figure 4B). No difference was found in CD8 T cell subpopulations between L1 and L2 (Figure 4C). Moreover, the ILCs/NK cell population from the lung samples also expressed CD45RA and CD56, and part of ILCs/NK cells expressed Granzyme B (Appendix A), but were comparable between groups.

A detailed analysis of the pulmonary myeloid cell population identified various subsets (Appendix A), including macrophages (cluster #1–4, CD206^+/−^CD68^+/−^CD163^+/−^), monocytes (cluster #6, 7, 16, CD14^+/−^CD16^+/−^), and dendritic cells (DC, cluster #5, 8–12, CD1c^+/−^CD14^+^/^−^CD123^+/−^). Significantly higher proportions of CD206^+^CD163^−^ macrophages (*p* = 0.0001) (Figure 4D, cluster #3) and decreased CD206^+^CD163^+^ macrophages (*p* = 0.0004) (Figure 4D, cluster #1) were found in L2 compared to L1. Comparing the marker profiles between these two macrophage subpopulations showed that CD206^+^CD163^+^ macrophages expressed CD16 and the activation markers CD38 and CD40, while CD206^+^CD163^−^ macrophages expressed CD11b and PD-L2 (Figure 4E). Moreover, proportions of CD14^+^CD1c^+^ monocyte-derived DCs (moDCs, Figure 4D, cluster #5) were decreased in L2 compared to L1 (*p* = 0.0012). In contrast, proportions of CD11b^+^CD123^+^ plasmacytoid DC (pDC, Figure 4D, cluster #8) were increased in L2 compared to L1 (*p* = 0.0009). The observed changes in myeloid subsets were due mostly to the surgery, whereas these subsets were similar in patients treated with BM-MSC vs. the placebo (Appendix A).

Altogether, our data demonstrate that LVRS induced recruitment of Treg and CD206^+^CD163^−^ macrophages to the lung, accompanied by alterations in DC subpopulations and decreased CD206^+^CD163^+^ macrophages, whereas BM-MSC treatment did not modulate this further.

### 3.3. Blood Monocytes May Contribute to Pulmonary Macrophage Populations

Since we observed alterations of myeloid cell subpopulations with distinct marker profiles in both PBMC and lung samples upon LVRS, we next explored the phenotype of myeloid cell subpopulations to determine potential migration, transition, and differentiation status in the blood and lungs. Firstly, we included CyTOF data from PBMC and lung-derived myeloid cells from L1 and L2, and performed an overview analysis in Cytosplore (Figure 5A). Different phenotypes of myeloid cell clusters were identified in both blood and lung samples based on the expression of markers, including various monocytes (cluster #1, 2, 10, 11), macrophages (cluster #3–6), and DC (cluster #7–9, 12, 13) (Figure 5B,C). CD14^+^ classical monocytes expressing CD141 could be identified as individual clusters in PBMC versus lung samples (Figure 5B,C, cluster 1 and cluster 2, respectively). The non-classical monocytes were similar in both PBMC and lung samples (Figure 5B,C, cluster #11). Moreover, subpopulations of macrophages (Figure 5B,C, cluster #6 CD206^+^CD163^+^; cluster #5 CD206^+^CD68^+^; cluster #4 CD206^+^; cluster #3 CD206^+^CD14^+^) were present in lung tissue and absent in PBMC samples. We identified three major CD1c^+^ DC subpopulations (Figure 5B,C). Firstly, the CD14^+^CD1c^+^ monocyte-derived DC (cluster #7) subpopulation expressing CD206 and CD14 was present in lung samples (Figure 5A,B). Second, CD11b^−^CD141^+^ DC1 (cluster #9) and CD11b^−^CD1c^+^ DC2 with CD206^+/−^CD14^+/−^ (cluster #8) phenotypes were identified in both PBMC and lung samples.

Finally, two subpopulations of pDC (CD11c^−^CD123^+^), which could be distinguished by CD11b or HLA-DR expression, were present in PBMC and lung samples (Figure 5B,C, cluster #12 and cluster #13, respectively), and were separated from all other myeloid clusters.

As we observed altered myeloid subpopulations upon L2 in both PBMC and lung samples, we determined potential migration, transitional, and differentiation states in the blood and lungs using pseudotime analysis, revealing cell trajectories by visualizing myeloid clusters in a diffusion map (Figure 5D and Appendix A). The separately clustered pDC clusters (Figure 5B, cluster #12 and cluster #13) were excluded from this analysis. The diffusion map shows that classical monocytes identified in PBMC branched into three subpopulations, including non-classical monocytes (CD16^+^), DCs (CD141^+^/CD1c^+^), and macrophages (CD206^+^), mainly present in the lungs (Figure 5E and Appendix A).

Taken together, the pseudotime analysis of the myeloid subpopulations suggests that the decreased number of circulating arterial blood-derived monocytes may contribute to various lung macrophage subpopulations.

## 4. Discussion

Here, we describe the alteration of the circulating and pulmonary immune composition in affected lung tissue from emphysema patients who underwent lung volume reduction surgery (LVRS) combined with BM-MSC or placebo treatment. Proportions of circulating lymphocytes were significantly increased and myelocytes significantly decreased at the second LVRS. More specifically, Foxp3^+^ Treg cells were significantly increased both in the circulation and in severe emphysematous lung tissue obtained at L2 compared to L1. The pulmonary CD206^+^CD163^−^ macrophages were significantly increased, whereas proportions of CD206^+^CD163^+^ macrophages and moDC were decreased in lung tissue obtained at L2 compared to L1. The pseudotime analysis revealed a potential migration of blood-derived monocyte subpopulations to severe emphysematous lung tissue, which subsequently may contribute to various macrophage subpopulations. However, no additional effects of BM-MSC on circulating and lung immune profiles compared to the placebo could be observed, suggesting that differences between immune cell profiles in tissue collected during the first and second surgery were in part resulting from the surgery itself.

The Cochrane Institute reported that the most effective treatment for emphysema is lung volume reduction surgery (LVRS) for patients with moderate to severe emphysema [3]. LVRS can result in improved inspiratory and expiratory capacity and may improve lung function by as much as 50%, and is associated with higher survival in patients with upper-lobe emphysema [4]. However, since the underlying pathological mechanisms driving inflammation and alveolar destruction continue in the absence of active cigarette smoking, the effect of LVRS may disappear after some 5 to 7 years post-surgery. Furthermore, results following bilateral LVRS on both lungs are variable due to varying patient characteristics [5,6], likely explaining the lack of effect on clinical parameters in our study.

MSC-based cell therapy offers a possible approach for treatment of chronic inflammatory diseases [26,27]. Previously, we conducted a phase I study with LVRS and autologous bone marrow-derived MSC (BM-MSC) treatment in patients with severe emphysema, where we found a 2-fold increase in CD3 T cells in alveolar septa 4 and 3 weeks after MSC infusion [13], while it was not clear whether this effect could not be attributed to BM-MSC administration or LVRS. In the present study, we included a placebo arm and determined the systemic and pulmonary immune cell profiles using high-dimensional CyTOF. There was no significant difference in clinical parameters, similar to a previous report [28,29]. These studies did show improvement in quality of life, increased FEV_1_ upon stratification for CRP levels, and improved walking distance [30], suggesting that MSC may be more potent in a pro-inflammatory environment and in clinically less stable patients, for example, patients who underwent recent exacerbations. This is supported by in vitro observations that cytoprotective and immunomodulatory MSC activities are modified upon exposure to serum from patients with hyper-inflammatory ARDS [8], indicating that the local microenvironment modulates the (anti-inflammatory and reparative) activity of MSC. These findings suggest that future studies using preconditioned MSC before administration, with, e.g., pro-inflammatory cytokines [9], may be explored further. This is further supported by studies in chronic inflammatory diseases, including ulcerative proctitis [10], where local MSC administration modulated the immune environment and improved disease activity.

Intravenous MSC administration, derived from various sources, either allogeneic or autologous, has been shown to be safe [31,32]. In contrast to our previously published pilot study [13], where we administered autologous BM-MSC, in the current study, we administered allogenic BM-MSC derived from healthy donors. This difference in MSC source between the two studies was needed due to alterations in the expansion protocol for BM-MSC, which resulted in an impaired expansion of BM-MSC from emphysema patients in the current study.

Until now, clinical trials studying the effects of MSC in COPD showed a limited clinical effect. This may be due to limited information on the optimal duration and frequency of MSC treatment, as well as timing of MSC infusion in the clinical course of the disease. Additionally, studies investigating the kinetics of the distribution of MSC, as well as inflammatory responses upon MSC infusion in short- and long-term periods, could provide more detailed information on the dynamics. Increased insight into the interaction of administered MSC with the lung microenvironment may have the potential to increase the efficacy of MSC treatment, e.g., by the preconditioning of MSC prior to infusion [9]. Furthermore, preclinical studies in which potential benefits of MSC treatment were reported applied relatively high doses of MSCs per kilogram of bodyweight [10], when compared to trials performed in humans, for which the dosing has been much lower because of safety and precaution reasons. Likewise, the route of administration, either systemically or locally, should be further investigated in future studies. More detailed information on the local responses in the unresected tissue to MSC treatment could also provide more information on the inflammatory and repair responses, but this is limited by the availability of lung tissue. Finally, isolated MSC populations may be heterogenous amongst donors and the source (e.g., bone marrow, adipose tissue, umbilical cord), but also subsets [11,12]. Although in the present study the chromosomal stability, expression of MSC-related markers [13], and proliferation capacity were monitored during and upon expansion, it is currently unknown whether additional testing of the anti-inflammatory, differentiation, and reparative capacity may be relevant prior to infusion, to select the most optimal MSC subset, source, or donor [12].

We performed a high-dimensional analysis of the circulating and pulmonary immune system and observed a significant increase in circulating and pulmonary Treg in L2 irrespective of BM-MSC treatment. The increase in proportions of immunomodulatory Treg may indicate the modulation of immune responses in emphysema [26]; however, we did not perform functional assays to assess this in the current study. In PBMC samples post L2, we found a higher proportion of CD8 Tem cells expressing Granzyme B, T-bet, and KLRG-1, which likely reflects a cytotoxic phenotype, and was reported to be related to mild to moderate COPD [33]. A detailed analysis of T cell subsets in lung tissue from COPD patients with mild to moderate emphysema [33,34] identified CD8 Tem and CD8 Tcm cells, which were shown to interact with myeloid and alveolar type II cells via interferon-γ production and thereby prevented tissue repair [35,36,37]. This was aligned with our previous study showing that cytotoxic CD8 T cells were abundant in emphysema patients and were able to produce interferon-γ [21].

In addition to altered circulating lymphocyte profiles, we identified significant changes in proportions of pulmonary myeloid cells collected at L2 compared to L1. CD206^+^CD163^+^ macrophages expressed the activation marker CD38 [38] and the costimulatory marker CD40, suggestive of a pro-inflammatory phenotype [39,40]. Proportions of these CD206^+^CD163^+^ macrophages were lowered at L2, whereas CD206^+^CD163^−^ macrophages expressing CD11b, which are shown to have immunosuppressive capacity [41], were upregulated at L2. The increase in proportions of anti-inflammatory myeloid subsets and lowering of pro-inflammatory subsets may indicate that a systemic immune response is modulated upon LVRS, although further research is needed to prove this hypothesis. A further analysis of monocytes and macrophages derived from the blood and lungs of emphysema patients using a pseudotime analysis visualized the transitional phenotypes in these subsets, and suggests that blood-derived monocytes potentially contribute to pulmonary macrophage subsets, which is in line with previous findings [40,42].

There are several limitations to this study, which may contribute to the observed limited effects of BM-MSC treatment. First, for both surgeries, the resected lung tissue was severely affected, and next to hampered ventilation, limited perfusion may have negatively affected the accessibility of circulating immune cells and BM-MSC. Second, we collected blood and lung samples at only 4 weeks after BM-MSC or placebo infusion and focused our study on the phenotypic cellular analysis of the circulating and pulmonary immune compartments. Future studies evaluating circulating inflammatory biomarkers **^30^** and pulmonary function [43] in shorter but also longer follow-up studies are needed for better understanding the effect of BM-MSC therapy in emphysema. Further, although no adverse effects of BM-MSC administration were found in this study, it is important to note that BM-MSC may present a heterogeneous population [11,12], and it is currently unknown which features and subpopulations are most relevant for the treatment of the various clinical phenotypes of emphysema. Furthermore, more translational studies are needed to optimize future clinical trials with MSC in emphysema, and potential late side effects of MSC treatment require monitoring [15]. Lastly, in this study, we have not yet visualized the tissue morphology and cellular components after BM-MSC/placebo infusion in situ. Spatial information may provide detailed insight on the localization of altered pulmonary (immune) cells and cellular interactions [21]. Future studies focusing on the localization of immune cells are needed to better understand the effect of BM-MSC treatment and LVRS and their role in progressive inflammation and (impaired) tissue repair in emphysema.

## 5. Conclusions

In conclusion, we confirm here that BM-MSC intravenous injection is safe, also when using allogeneic BM-MSC. Our study did not show effects of BM-MSC on immune profiles present in resected lung tissue of patients with severe emphysema. However, we showed that systemic as well as pulmonary immune profiles were altered upon LVRS. We propose that these alterations may be beneficial in view of their anti-inflammatory signature, which may be favorable for endogenous repair processes in emphysema.

## Figures and Tables

**Figure 1 cells-13-01636-f001:**
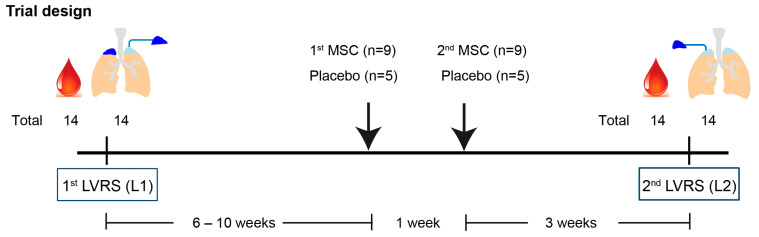
Trial design.

**Figure 2 cells-13-01636-f002:**
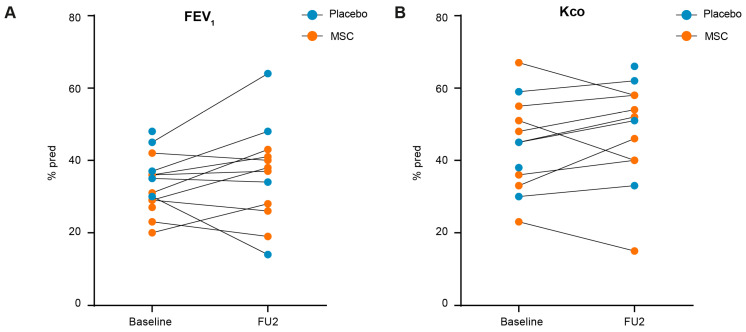
Clinical parameters. (**A**) Forced expiratory volume in 1 s (FEV_1_) and (**B**) diffusing capacity of the lungs for carbon monoxide (K_co_) were evaluated at screening prior to surgery (baseline) and at follow-up 1 year after L2 (FU2) in patients that received BM-MSC (n = 8, orange) or a placebo (n = 4, blue). One patient in the control group and one patient in the BM-MSC group did not undergo the second lung function measurement.

**Figure 3 cells-13-01636-f003:**
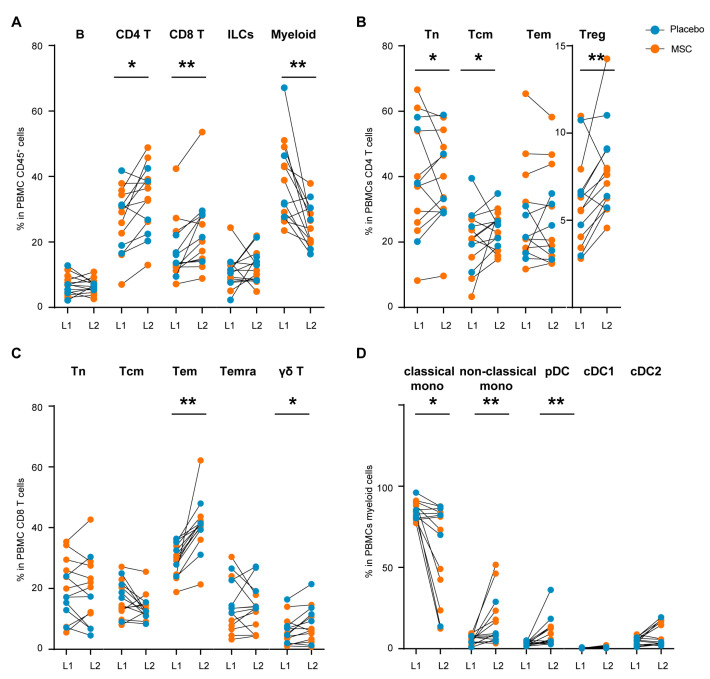
Comparison of major immune cell lineages and subpopulations in blood following LVRS and BM-MSC/placebo treatment. (**A**) CyTOF was used to determine the percentage of major immune lineages among all CD45+ cells (9 × 10^6^ cells) in peripheral blood mononuclear cells (PBMCs) in emphysema patients at L1 and L2, and receiving BM-MSC (n = 9, orange) or placebo (n = 5, blue) treatment between the surgeries. A detailed analysis was performed to determine (**B**) the percentage of CD4 T cell subpopulations (2.8 × 10^6^ cells), (**C**) the percentage of CD8 T cell subpopulations (1.9 × 10^6^ cells), and (**D**) the percentage of myeloid cell subpopulations (1.3 × 10^6^ cells). * *p* < 0.05 and ** *p* < 0.01 by the Wilcoxon matched-pair signed rank test. Tn, CCR7^+^CD45RO^−^ naïve T cells; Tcm, CCR7^+^CD45RO^+^ central memory T cells; Tem, CCR7^−^CD45RO^+^ effector memory T cells; Temra, CCR7^−^CD45RA^+^ terminally differentiated T cells; Treg, Foxp3^+^ regulatory T cells; γδ T, γδ^+^ T cells; classical mono, CD14^+^ classical monocytes; non-classical mono, CD16^+^ non-classical monocytes; pDC, CD123^+^ plasmacytoid dendritic cells; cDC1, CD141^+^ dendritic cells; cDC2, CD1c^+^ dendritic cells.

**Figure 4 cells-13-01636-f004:**
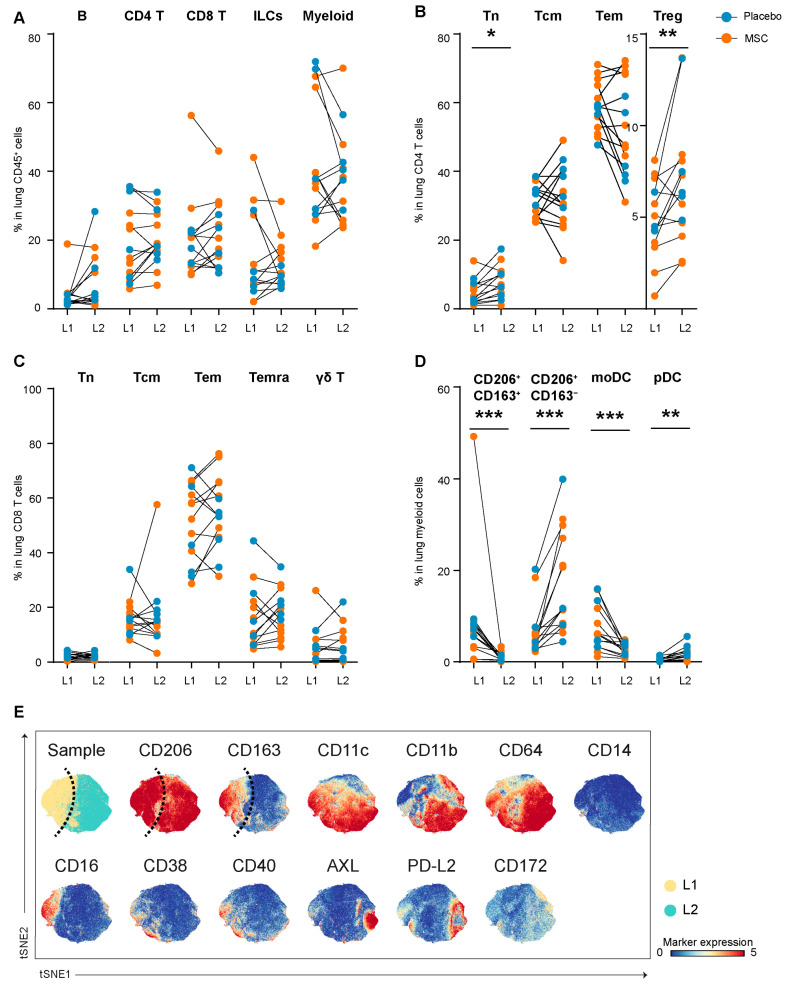
The comparison of the major pulmonary immune lineages and subpopulations following LVRS and BM-MSC/placebo treatment. (**A**) CyTOF measurements were performed on resected lung tissue from L1 and L2 in emphysema patients that received BM-MSC (n = 9, orange) or a placebo (n = 5, blue), and percentages of major immune lineages among all CD45^+^ immune cells (6.2 × 10^6^ cells) were determined. The detailed analysis of the major immune lineages identified (**B**) the percentage of CD4 T cell subpopulations (1.4 × 10^6^ cells), (**C**) the percentage of CD8 T cell subpopulations (1.3 × 10^6^ cells), and (**D**) the percentage of myeloid cell subpopulations (1.5 × 10^6^ cells). (**E**) Myeloid cell subsets in the lungs at L1 (yellow) and L2 (teal) were analyzed in detail by visualizing the marker expression profiles of CD206^+^CD163^+^ and CD206^+^CD163^−^ subpopulations; these subsets are indicated with dashed lines (red indicates high expression, blue indicates low/absent expression). * *p* < 0.05, ** *p* < 0.01, and *** *p* < 0.001 by the Wilcoxon matched-pair signed rank test. Tn, CCR7^+^CD45RO^−^ naïve T cells; Tcm, CCR7^+^CD45RO^+^ central memory T cells; Tem, CCR7^−^CD45RO^+^ effector memory T cells; Temra, CCR7^−^CD45RA^+^ terminally differentiated T cells; Treg, Foxp3^+^ regulatory T cells; γδ T, TCR γδ^+^ T cells; pDC, CD123^+^ plasmacytoid dendritic cells; moDC, CD1c^+^CD141^+^ monocyte-like dendritic cells.

**Figure 5 cells-13-01636-f005:**
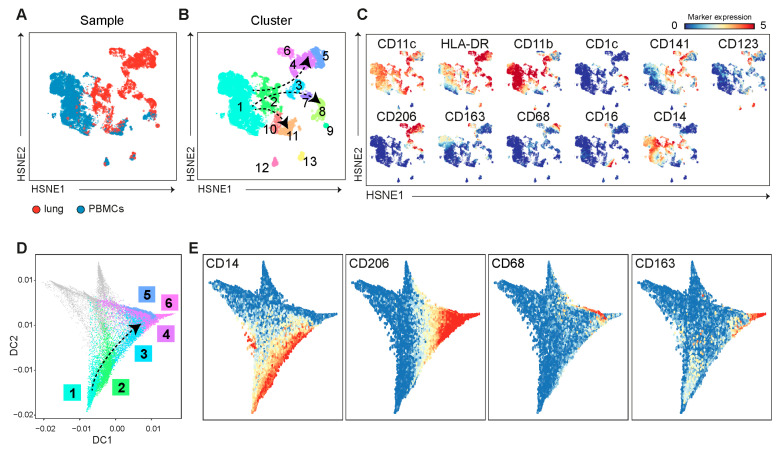
The pseudotime analysis on the PBMC and lung-derived myeloid cells suggests that blood monocytes may contribute to pulmonary macrophage populations. (**A**) A collective HSNE analysis was performed on PBMCs (blue) and lung-derived (red) myeloid cell populations at L1 and L2, and (**B**) identified clusters amongst the myeloid cells, indicated with colors and numbers, based on (**C**) expression profiles of myeloid markers (red indicates high expression, blue indicates low/absent expression). (**D**) The trajectory of myeloid cells was estimated using the pseudotime analysis based on the marker expression and is shown in a diffusion map, with cell clusters that are identified in (**B**). (**E**) The expression of myeloid markers along the different trajectories is visualized.

**Table 1 cells-13-01636-t001:** Baseline characteristics of emphysema patients.

Baseline Characteristics	Mean (±SEM)
Age (years)	56 ± 1.3
Body mass index (kg/m^2^)	25.2 ± 0.9
FEV_1_ (% pred)	32.7 ± 3.1
K_co_ (% pred)	42.6 ± 3.7
RV/TLC (%)	59.2 ± 1.9
FEV_1_/FVC (%)	54 ± 5
PD 15 left lung (Houdsfield Units)	960 ± 3.7
PD 15 right lung (Houdsfield Units)	964 ± 3.6
Cardiac ejection fraction (%)	67 ± 1.5
End-diastolic volume (ml)	66 ± 3.6
Smoking history (pack-years)	23 ± 4.5

Overview of characteristics of emphysema patients at screening prior to surgery (baseline). Definition of abbreviations: FEV_1_ = forced expiratory volume in 1 s; K_co_ = diffusing capacity of the lungs for carbon monoxide; RV/TLC = residual volume/total lung capacity; PD 15 left/right lung (HU) = 15th percentile density (PD15) measured as Hounsfield units (HU) in left and right lung.

## Data Availability

The original contributions presented in the study are included in the article/Appendix A; further inquiries can be directed to the corresponding author.

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
