# Peer review of "Pulmonary and Systemic Immune Profiles Following Lung Volume Reduction Surgery and Allogeneic Mesenchymal Stromal Cell Treatment in Emphysema"

_cells, 2024, doi:10.3390/cells13191636_

Round 1

Reviewer 1 Report

Comments and Suggestions for Authors

There are few human translational studies performed in stem cell research in patients with chronic obstructive pulmonary disease (COPD) and/or pulmonary emphysema. In the future, stem cell therapy could be a treatment for this incurable disease but as of now, stem cell therapy is still to be considered as an area of active research. In this paper, Jia L and coworkers studied the pulmonary and systemic immune profiles following lung volume reduction surgery and allogeneic mesenchymal stromal cell treatment in emphysema. This work is a placebo-randomized trial well-designed and conducted. This working group has already published a phase I study for intravenous autologous mesenchymal stromal cell administration to patients with severe emphysema. The manuscript is clear and well written and I have only minor comments.

-          In the results section authors reported that clinical parameters were unaltered upon LVRS and BM-MSC/ placebo treatment. It would be appropriate to comment in the discussion why no change in lung function was obtained after surgical reduction of emphysema and to better describe the type of surgery that was performed by comparing the results with those obtained by other groups before and after LVRS alone.

-          The lack of clinical response after allogeneic stem cell infusion should be deepened in discussion and hypotheses made as to why the lack of response;

-          Table n.1 does not read well should be improved for easier reading and Tiffenau's index should be added among the functional parameters.

-          This is a negative study in which no major results were found after i.v. infusion of allogeneic mesenchymal stromal cells. It should be further investigated why there is a lack of response by formulating hypotheses and paving the way on how to conduct future studies that can better investigate the efficacy of this treatment.

-          In the end although no adverse effects were found in this study it must be emphasized in the discussion that although stem cells would be likely to represent a heterogeneous population of cells, the different cell subsets and their importance in the pathogenesis of the different clinical phenotypes of emphysema need to be fully characterized before progressing to clinical trials. Moreover, the potential side effects of stem cell therapy are underestimated. We should not ignore that some of the most deadly neoplasms arise from stem cells (see reference:DOI: 10.1080/15412555.2018.1536116).

Reviewer 2 Report

Comments and Suggestions for Authors

The authors performed a Pulmonary and systemic immune profile using top-of-the-art technology to characterize the phenotypic cellular changes induced in lung volume reduction surgery followed by allogeneic mesenchymal stromal cell treatment in emphysema patients.

The significant outcomes of this study include an increased profile of CD4 and CD8 T Lymphocytes, a decrease in myeloid cells, and a predominant anti-inflammatory phenotype after the lung volume reduction surgery, independent of the allogeneic mesenchymal stromal cell treatment. 

The overall study design, methodology, and results support the author's claims. The manuscript is well organized and presented and will be accepted for publication after minor revisions:

1. There is an inconsistency in Figure 2 and some of the panels of Figure 3. Although the methods and figure legend state that there are BM-MSC (n=9, orange) or placebo (n=5, blue), in the actual figure, it seems that there are only n=8 and n=4, respectively.

2. It would be good to include in the Discussion the results observed in the context of the immune response over the time between the two sample collections. Similar to the proposed additional studies at shorter and longer times after the mesenchymal stromal cell treatment, the cellular immune profile would be dynamic over the >10-week period between L1 and L2 collections. We can assume that a more proinflammatory profile will be observed early (around four weeks).  
